# Isotopic and Chemical Assessment of the Dynamics of Methane Sources and Microbial Cycling during Early Development of an Oil Sands Pit Lake

**DOI:** 10.3390/microorganisms9122509

**Published:** 2021-12-03

**Authors:** Greg F. Slater, Corey A. Goad, Matthew B. J. Lindsay, Kevin G. Mumford, Tara E. Colenbrander Nelson, Allyson L. Brady, Gerdhard L. Jessen, Lesley A. Warren

**Affiliations:** 1School of Earth, Environment and Society, McMaster University, Hamilton, ON L8S 4K1, Canada; coreygoad74@gmail.com (C.A.G.); bradyal@mcmaster.ca (A.L.B.); 2Department of Geological Sciences, University of Saskatchewan, Saskatoon, SK S7N 5E2, Canada; matt.lindsay@usask.ca; 3Department of Civil Engineering, Queen’s University, Kingston, ON K7L 3N6, Canada; kevin.mumford@queensu.ca; 4Department of Civil and Mineral Engineering, University of Toronto, Toronto, ON M5S 1A4, Canada; tara.nelson@utoronto.ca (T.E.C.N.); lesley.warren@utoronto.ca (L.A.W.); 5Instituto de Ciencias Marinas y Limnologicas, Faculated de Ciencias, Universidad Austral de Chile, Valdivia 5090000, Chile; gerdhard.jessen@uach.cl

**Keywords:** pit lakes, oil sands, tailing reclamation, methane, methanotrophy, phospholipid fatty acids, carbon isotopes, ebullition

## Abstract

Water-capped tailings technology (WCTT) is a key component of the reclamation strategies in the Athabasca oil sands region (AOSR) of northeastern Alberta, Canada. The release of microbial methane from tailings emplaced within oil sands pit lakes, and its subsequent microbial oxidation, could inhibit the development of persistent oxygen concentrations within the water column, which are critical to the success of this reclamation approach. Here, we describe the results of a four-year (2015–2018) chemical and isotopic (δ^13^C) investigation into the dynamics of microbial methane cycling within Base Mine Lake (BML), the first full-scale pit lake commissioned in the AOSR. Overall, the water-column methane concentrations decreased over the course of the study, though this was dynamic both seasonally and annually. Phospholipid fatty acid (PLFA) distributions and δ^13^C demonstrated that dissolved methane, primarily input via fluid fine tailings (FFT) porewater advection, was oxidized by the water column microbial community at all sampling times. Modeling and under-ice observations indicated that the dissolution of methane from bubbles during ebullition, or when trapped beneath ice, was also an important source of dissolved methane. The addition of alum to BML in the fall of 2016 impacted the microbial cycling in BML, leading to decreased methane oxidation rates, the short-term dominance of a phototrophic community, and longer-term shifts in the microbial community metabolism. Overall, our results highlight a need to understand the dynamic nature of these microbial communities and the impact of perturbations on the associated biogeochemical cycling within oil sands pit lakes.

## 1. Introduction

Bitumen extraction from mined oil sands in the Athabasca oil sands region (AOSR) of northeastern Alberta, Canada, generates large volumes of fluid fine tailings (FTT). Various approaches are being developed and implemented to divert FFT from tailings ponds to permanent reclamation landforms [1]. Water-capped tailings technology (WCTT) involves the permanent storage of FFT under a shallow water column cap, creating pit lakes from exhausted mine pits [2]. Commissioned in 2012 by Syncrude Canada Ltd., Base Mine Lake (BML) was the first full-scale demonstration of WCTT established in the AOSR. Approximately 23 of these aquatic reclamation landforms are currently planned for the AOSR, and 8 of these will contain tailings [2]. Ultimately, the long-term success of pit lakes as reclamation landforms is contingent on the effective sequestration of tailings below the water cap, and on the development of conditions that support ecological function as part of a boreal forest ecosystem [3].

Sustained dissolved oxygen concentrations within the water cap are critical to the long-term viability of pit lakes. Dissolved oxygen is also essential for aerobic respiration and, therefore, the development of a functioning aquatic ecosystem. In addition, oxygen supports the microbial degradation of potentially hazardous organic compounds, such as naphthenic acids (NAs), present in the oil sands' process-affected water and associated with fluid tailings [4]. However, the oxygen supplied by physical and biological processes can be microbially depleted via reactions with methane and other oxygen-consuming constituents released from fluid tailings [5,6,7]. On the basis of observations from tailing ponds and laboratory studies, microbial methanogenesis is presumed to be continuing to produce methane within the FFT underlying BML [8,9,10,11,12,13] (Figure 1). This methane can be transported to the overlying water cap through three main processes: (1) Advective transport during FFT settlement and dewatering; (2) Diffusive transport across concentration gradients; and (3) The dissolution of gaseous methane from bubbles migrating vertically from FFT through the water cap and released at the lake surface (Figure 1). Once in the oxic water column, this methane can be consumed via the action of aerobic methanotrophic microbes, or, alternatively, it may partition out of the water column at the air–water interface. The same transport processes can also deliver organic carbon constituents (e.g., residual petroleum hydrocarbons) from the FFT to the water column that may be consumed by aerobic heterotrophic microbes, alongside the heterotrophic consumption of any photosynthetically produced dissolved organic carbon (DOC). Indeed, Risacher et al. (2018), and Arriaga et al. (2019) found that the oxygen concentrations and distribution in the BML water column, during early time points in lake development (2015–2016), were correlated with methane concentrations and the rates of the downward physical mixing of the oxygen supply and depth-dependent consumption. Their results imply that FFT-associated microbial methane production, the mobilization to, and the subsequent oxidation within the water cap, was playing an important role in influencing the oxygen concentrations.

These previous studies highlight the influence of methane on the BML oxygen concentrations and distribution and indicate the need for a greater understanding of the seasonal changes in dissolved methane concentrations, in the role of microbial methane oxidation relative to other carbon and energy sources available to the BML microbial community, and how these factors may have changed over the longer-term BML development trajectory. Such an understanding is important to enabling the assessment and prediction of the lake development, which will inform the sound design and management of future applications of WCTT in the AOSR.

This study used a combination of phospholipid fatty acid (PLFA) and a stable isotopic analysis of methane and PLFA to investigate the methane dynamics within BML. Analysis of the stable isotopic composition of methane, that is, the ratio of heavy-to-light isotopes of carbon (δ^13^C) and hydrogen (δ^2^H), is a well-established tool that can identify the sources and microbial oxidation of methane in environmental systems [14]. The ranges of δ^13^C and δ^2^H can elucidate the methane production pathways [14], while trends of the isotopic fractionation of methane to values more enriched in the heavier isotopes of carbon can identify the occurrence of methane oxidation [15].

PLFA are the fatty acid components of membrane lipids produced by bacteria and eukarya [16], but not by archaea, such as methanogens, which produce predominantly ether-bound membrane lipids [17]. Total PLFA concentrations can be used, in conjunction with conversion factors, to estimate the abundances of the bacterial and eukaryotic components of environmental microbial communities [18], while the distributions of individual PLFA can provide insight into the microbial community structure [16,19]. In some cases, individual PLFA can be indicative of specific types of organisms, for instance, C16:1ω5t is considered to be a biomarker for Type I and X methanotrophs [20]. In addition, PLFA analysis can be combined with isotopic analysis such that the δ^13^C of PLFA can be used to elucidate the microbial carbon sources and metabolic pathways [21].

The objectives of this study were to investigate the methane dynamics within BML over four years (2015–2019), including data from under the ice and after the addition of alum (fall 2016) to increase water clarity. Phospholipid fatty acid (PLFA) abundances, biomarker distributions, and stable isotopic compositions were used to identify the occurrence and importance of methane oxidation to the microbial community within the epilimnion and hypolimnion of BML over time. Methane concentrations and isotopic compositions, in conjunction with mass balance modeling, were used to evaluate trends in the methane sources and the extent to which water-cap-associated microbial oxidation was removing the dissolved methane. The addition of alum to BML in the fall of 2016 allowed the study to include a characterization of the microbial community response to a large-scale management perturbation. The results of this study will inform not only the ongoing adaptive management of BML, but also development and management practices for future applications of WCTT.

## 2. Materials and Methods

### 2.1. Site Description

Base Mine Lake is located at the Mildred Lake Mine, approximately 35 km north of Fort McMurray, Alberta, Canada (Figure 2). This reclamation landform covers 7.8 km^2^ and initially contained 186 Mm^3^ of FFT, which were deposited into the former West In-Pit between 1994 and 2012. Following commissioning, in December 2012, BML contained up to 48 m of FFT, stored under a 5-m water cap. Approximately 20 Mm^3^ of freshwater was added from the adjacent Beaver Creek Reservoir (BCR) to raise the final water surface elevation to 308 m above sea level [22]. This freshwater addition increased the water-cap depth from approximately 5 to 7.5 m. In situ FFT settlement and dewatering between 2013 and 2018 subsequently increased the water-cap depth to between 8 and 12 m, with an average depth of approximately 10 m. Over the course of the study, varying volumes of surface water were removed from BML for re-use in the bitumen extraction process; this water was replaced with freshwater from BCR to maintain the final surface water elevation. Freshwater inputs, combined with in situ microbial processes, are expected to support long-term improvements in the surface water quality, thereby mitigating the potential impacts of the dissolved constituents contributed from the underling FFT. 

Since monitoring began in 2013, BML consistently turns over in the spring and fall and remains thermally stratified between these events. The temperature profiles collected during our study (2015–2018) reveal that the epilimnetic layer occurred between 0 to 5 m below the water surface (mbws), the metalimnion between 5 and 6 mbws, and the hypolimnion from 6 mbws to the FFT–water interface (FWI), which was initially located at approximately 8 mbws, and increased to approximately 12 mbws over the course of the study [7]. Risacher et al. (2018) report that completely anoxic conditions did not develop in the BML water column during their study period (i.e., May to September of 2015 and 2016). These authors observed consistent dissolved oxygen (DO) depth profiles during their 2015 and 2016 sampling campaigns. They reported maximum O_2_ saturations of 75 to 80% in the upper epilimnion (0–5 mbws), followed by a sharp decline through the metalimnion (5–6 mbws) to ~30% saturation, and a continued rapid decline to ~1 to 2% saturation in the hypolimnion (~6 mbws to FWI). Risacher et al. (2018) also observed that the water-column mixing associated with the fall turnover (late August–early September) increased the O_2_ saturation to 30% through the lower water cap in 2015 and 2016.

As part of the adaptive management of BML, alum addition was undertaken in the fall of 2016 in order to decrease the water-cap turbidity. The monitoring results indicate that the addition of alum led to very low turbidity during winter 2016/2017, and to ongoing lower levels of turbidity since that time [23].

### 2.2. Sampling Locations and Frequency

Depth-dependent water column samples were collected every 2 to 4 weeks over the following periods: May to September 2015, July to September 2016 (n.b., site access in May and June 2016 was prevented because of wildfires), May to October 2017, and May to August 2018. Additional samples were collected from under the ice in February 2017, and in February and March 2018. The physicochemical profiles of the BML water column were collected concurrent with all sampling times. Sample locations were consistent with those previously reported by Risacher et al. (2018). During the 2015 campaign, samples were collected at three permanently anchored floating platforms (P1, P2, P3) along a diagonal transect from the southwest (P3) to northeast (P2), with P1 positioned near the center of BML (Figure 2). In subsequent years, water-column samples were taken along a higher-resolution depth profile only at P1 because the data was consistent between all three platforms in 2015. A specialized sampling boat was used to collect samples from May to October of each year. Samples were collected in February 2017, February 2018, and March 2018 through an augered hole in the ice adjacent to P1. In 2016, water-column and FFT samples were also collected by a fixed interval sampler, described in Dompierre et al. (2016). Profiles of the physiochemical characteristics (pH, temperature, dissolved oxygen concentration and % saturation, specific conductivity, ORP, turbidity, and salinity) were measured from the BML surface to the FFT–water interface (FWI) at ~50 cm intervals, using a YSI Professional Plus 6-Series Sonde (YSI Incorporated), as per Risacher et al. (2018), and Arriaga et al. (2019). These profiles were used to identify the sampling depths for each sampling campaign. For all sampling campaigns (141 over the 2015–2018 sampling period), water samples were collected from a minimum of three depths to a maximum of twelve depths, and always included sampling at the FWI. The samples used in this study were selected from this larger sample set.

### 2.3. Sample Collection and Preservation

#### 2.3.1. Water-Column Sampling

At each sampling depth, water samples were collected for the dissolved methane concentration [CH_4(aq)_] and stable isotope (δ^13^C, δ^2^H) analyses into an air-tight Van Dorn bottle (Water Mark, Forestry Suppliers, Jackson, MS, USA, or Beta Water Sampler, Wildco, Yulee, FL, USA). Immediately upon retrieval, the water samples were withdrawn from the Van Dorn into air-tight 60-mL syringes and were gently injected through stainless steel needles into pretreated 60-mL glass serum bottles (Wheaton Scientific Products, Millville, NJ, USA). Paired water samples were collected from the same Van Dorn sampling bottle for the methane concentration (30 mL water, leaving 30 mL headspace in the 60-mL serum bottle) and isotopic characterization (50 mL water, leaving 10 mL of headspace in the 60-mL serum bottle). Serum bottles, which contained 3.7 mg of saturated mercuric chloride solution, were plugged with 13-mm blue butyl rubber septum stoppers, and were crimp-sealed with an aluminum ring prior to evacuation by vacuum pump [24]. All sample bottles were stored inverted to prevent gas exchange or sample loss through the septum stoppers. 

#### 2.3.2. Fixed Interval FFT Sampling

Samples of FFT for dissolved CH_4_ analyses were collected from discrete depths below the FWI using either a 250-mL fixed interval sampler (see Dompierre et al., 2017), or a 4-L fluid sampler (see Dompierre et al., 2016). Immediately upon retrieval of the sampler, FFT were transferred into 250-mL high-density polyethylene bottles and immediately sub-sampled using 60-mL polyethylene catheter syringes. These samples were then gently extruded into 120-mL amber glass serum bottles such that there was a 60-mL headspace, sealed with 13-mm blue butyl septum stoppers, and crimp-sealed with an aluminum ring. It was not possible to chemically sterilize the FFT in the field, so the samples were immediately frozen (–20 °C) to inhibit methanogenesis. The samples were transferred to a refrigerator (4 °C) seven days before analysis to allow the methane in the FFT to equilibrate with the headspace. After analysis, the bottles were opened to determine the FFT water content and headspace volume. Air blanks were collected during sampling following the same methods described for the FFT samples. To demonstrate the effectiveness of this method, and to obtain a closely spaced series of samples, one set of water samples was collected from the water column using the FIS system. 

#### 2.3.3. PLFA Sampling

Water samples for phospholipid fatty acid (PLFA) analysis were collected with the air-tight Van Dorn bottle (Water Mark, Forestry Suppliers, Jackson, MS, USA). Upon retrieval of the Van Dorn from depth, 1 L of sample water was withdrawn into a 1-L Nalgene bottle prerinsed with methanol (MeOH). Water samples were frozen at the end of each sampling day and stored on-site and, subsequently, at the laboratory at McMaster University at –20 °C until analysis. 

### 2.4. Analytical Methods

#### 2.4.1. Dissolved Methane (CH_4(aq)_) Concentration Analysis

Sample bottles under residual vacuum were brought to atmospheric pressure prior to analysis by adding circa 25 mL of high-purity helium (AlphaGaz 99.999%) until the pressure within the bottle was equalized with the atmosphere. The [CH_4(aq)_] in the water column and the FFT samples were quantified by injecting 1000 μL of sample bottle headspace on an SRI 8610C gas chromatograph (GC; silica gel column, 0.91 m × 2.1 mm) coupled to a thermal conductivity detector (TCD) and a flame ionization detector (FID) with an oven temperature hold at 40 °C. All samples were injected in triplicate (RSD ≤ 10%) and were quantified on methane calibration curves created and maintained directly from gas tanks with known methane concentrations. Peak integration and quantification were conducted using PeakSimple v3.29 (SRI Instruments, Torrance, CA, USA). The [CH_4(aq)_] were calculated from the total headspace methane mass, which was determined from the headspace concentration and volume, assuming all the methane had partitioned into the headspace, divided by the original water volume in the water column or FFT sample. 

Laboratory air blanks were consistently below detection, and field air blanks yielded concentrations of ≤0.1 μM. The trace amount of methane in the field air blanks was likely related to methane ebullition from the lake surface around the sampling boat. Every effort was made to exclude air during sample collection, so these field air blanks represent a worst-case scenario of air contamination, and laboratory blanks indicate that the field samples received negligible contributions of atmospheric methane.

#### 2.4.2. Dissolved Methane Stable Isotope Analysis

Stable carbon isotope analysis of methane, δ^13^C-CH_4(aq),_ was achieved by injecting 300 μL of headspace gas (in triplicate, RSD ≤ 10%) on an Agilent 6890 GC (GSQ; 30 m × 0.32 mm) coupled to a Thermo Delta Plus XP Isotope Ratio Mass Spectrometer via a Conflo III interface (GC-IRMS), with a continuous oven temperature hold at 30 °C. The GC-IRMS data was processed with Isodat NT 2.0 software (©Thermo Electron Company, Waltham, MA, USA). Carbon isotope ratios (δ^13^C) were normalized to the Vienna Peedee belmenite (VPDB) standard. All samples were injected in triplicate, with a δ^13^C standard deviation of approximately ± 0.5‰, based on standard reproducibility and instrument accuracy. In order to obtain accurate and reproducible isotope analysis results, the concentration of methane in the sample had to be greater than 20 μM, which limited this approach to the hypolimnion, FFT porewater, and spring 2017 samples.

Deuterium analysis of methane, δ^2^H-CH_4(aq),_ was carried out at Memorial University, in Newfoundland, on an Agilent 6890N GC coupled to a Delta V Plus IRMS via a GC Combustion III interface (30 m × 0.32 mm × 15 μm, Carboxen 1010 column) with an oven temperature program of 110 °C for 5.5 min, to 180 °C at 35 °C/min, and a final hold for 2 min. The temperature conversion reactor was held at 1450 °C. Hydrogen isotope ratios were also normalized to the VPDB standard. On the basis of the standard reproducibility and instrument accuracy, the δ^2^H measurement error is approximately ± 5.0‰. Because of the concentration requirements for δ^2^H analysis, which are greater than those for δ^13^C analysis, the δ^2^H-CH_4(aq)_ could only be measured for FFT porewater samples.

### 2.5. PLFA Identification and Quantification 

Water samples for PLFA analysis were thawed at 4 °C, at McMaster University, prior to gravity filtration through precombusted 0.7-μm glass fiber filters, followed by methanol-rinsed 0.45-μm polyvinylidene fluoride (PVDF) filters to collect the pelagic biomass. Filters were freeze-dried for 48–72 h prior to solvent extraction. PLFA were extracted and isolated from the sample filters using a modified Bligh and Dyer procedure [25,26]. In brief, extraction used a mixture of 2:1:0.8 methanol:dichloromethane:phosphate buffer. Extractions were sonicated and agitated overnight at room temperature before undergoing phase separation. Phospholipids were separated from the total lipid extract (TLE) by silica gel chromatography using three fractions (F1 = DCM, F2 = acetone, F3 = MeOH). The phospholipids isolated in the F3 fraction were evaporated to dryness, redissolved with KOH and a 1:1 toluene:methanol mixture (using methanol with known δ^13^C), then heated to 37 °C for an hour to facilitate the conversion of phospholipids to fatty acid methyl esters (FAMEs ) for analysis by gas chromatography-mass spectrometry (GC-MS). The FAMEs were purified via secondary silica gel chromatography (F1 = 4:1 hexane:DCM, F2 = DCM containing PLFA as FAMEs, F3 = MeOH) prior to analysis by GC-MS. 

All PLFA samples were identified and quantified by mass analysis on an Agilent 6890 GC (Column: DB-5MS, 0.25-μm film thickness, 30-m length, 0.32-mm ID) connected to an Agilent 5973 quadrupole mass spectrometer. The oven ramp temperature program was as follows: 50 °C for 1 min, increasing by 20 °C/min up to 130 °C, then 4 °C/min to 160 °C, and 8 °C/min to 300 °C, and a hold for 5 min. MSD Chemstation (Agilent Technologies, Santa Clara, CA, USA) was used to identify the FAMEs through the database-matching of spectra and overlaid spectra comparisons to two reference standards (Matreya PLFA mix, and Supelco 37 FAME mix). The quantification of FAMEs was based on six-point calibration curves generated for four different FAMEs (14:0, 16:0, 18:0, and 20:0; R^2^ > 0.99), back-calculating using instrument areas and integration tools within MSD Chemstation. 

### 2.6. PLFA Stable Carbon Isotope Analysis 

The stable carbon isotopes of the PLFA samples were analyzed using an Agilent 6890 GC (Column: Agilent: DB-5, 0.25-μm film thickness, 30-m length, 0.32-mm ID) attached to a Conflo III Combustion Interface, followed by a Thermo Delta Plus XP IRMS (GC-C-IRMS). Isodat NT 2.0 software (Thermo Electron Company, USA) was used to analyze the GC-C-IRMS results. Carbon isotope ratios were normalized to the VPDB standard. A δ^13^C value correction was applied to account for the addition of a methyl group from the isotopically characterized methanolic KOH during FAME derivatization. For FAMEs that were not well baseline-separated, particularly the unsaturates, the pooled δ^13^C values were generated by integrating across the entirety of the co-eluting peak [27,28,29]. All samples were injected in triplicate, with a δ^13^C standard deviation of approximately ± 0.5‰, based on standard reproducibility and instrument accuracy. 

### 2.7. Calculations of Methane Mass Loading from Potential Sources

A first-order assessment of the processes contributing to the dissolved methane to the BML water column was performed using the water column and FFT [CH_4(aq)_] from 2016, the only year a complete dataset including the FFT porewater methane concentrations was available. During this year, the hypolimnetic aqueous methane concentrations varied from 80 to 150 uM (Figure 3), while the dissolved methane concentrations in the upper FFT varied from 1000 uM to 3770 uM (Figure 4). The annual mass loading of dissolved methane by molecular diffusion across the FWI was calculated using Fick’s law, assuming a methane diffusivity of 1.49 × 10^−5^ cm^2^ s^−1^, the FFT surface area of 6.31 × 10^6^ m^2^, and the maximum (3770 to 80 uM over the upper 10 cm) and minimum (1000 to 150 uM over 50 cm) concentration gradients from the upper FFT to the hypolimnion bottom water. While the solids content was not considered in this estimate, the low solids content of upper FFT (<25%) would correspond to a relatively small increase in diffusive mass loadings. The same range of methane concentrations was used to estimate the annual input of dissolved methane from advective water expression associated with FFT settlement, which averaged 0.73 m a^−1^ from May 2013 to October 2015 [6]. The maximum advective mass loading was a calculated dissolved methane concentration of 4000 μM, which was generally consistent with the highest observed concentrations in upper FFT porewater (~1.5 m depth), and with calculated methane solubilities using the thermodynamic model of Duan and Mao (2006) for the highest in situ temperatures of 14 °C [30]. The minimum advective mass loading was calculated assuming the lowest observed methane concentrations in the upper FFT (1000 μM within the upper 50 cm). These are simplified endmember calculations, since there is evidence that FFT porewater methane concentrations increase with depth from well below saturation, at 10 cm below the FWI, to saturation at 0.5–2 m. However, the use of these maximum and minimum values enabled the mass loading from both advection and diffusion to be bounded to a first order. 

The mass loading, related to methane dissolution from bubbles rising through the water column, was determined using the model of Leifer and Patro (2002) [31]. This model simulates the mass transfer of multiple gas components between the water and a single rising gas bubble. The details of the modeling are presented in the "Appendix A" section. Leifer and Patro (2002) [31] reviewed several expressions for estimating the bubble-rise velocity. Here, we used their expression for dirty bubbles (i.e., those with a gas–water interface contaminated by surfactant), with 0.6 mm < *r* < 10 mm. This equation predicted bubble velocities of 0.15–0.23 m s^−1^, which is in good agreement with the bubble velocities in BML, measured by sonar, of 0.19–0.26 m s^−1^. The mass transfer of each component in the gas phase across the bubble interface is described by diffusion through a thin stagnant film, combined with Henry’s Law and Dalton’s Law, and assuming an interfacial area for a spherical bubble (Leifer and Patro, 2002) [31]. The approach was verified against the simulations of a methane bubble with a radius of 2.5 mm rising in 5 m of water containing dissolved oxygen and nitrogen (Liefer and Patro, 2002) [31]. 

In this study, the model was implemented assuming that the bubbles released from the FFT were composed either entirely of methane, or of 75% methane and 25% nitrogen (by volume), based on the results of the preliminary gas analyses (unpublished data), and allowed to exchange mass with a water column that contained dissolved oxygen, nitrogen, and methane. The concentrations of oxygen and methane in the water column varied with the depth and were based on observations in this study and those reported by Risacher et al. (2018). The dissolved nitrogen concentration was assumed to be in equilibrium with the atmosphere over the entire depth of the water column. The water temperature also varied with the depth and was based on observations in this study. The results of these simulations are sensitive to the initial radius of the bubble released from the FFT, which is unknown. Therefore, initial bubble radii of 1, 3, and 5 mm were simulated. Although the equations presented in "Appendix A" simulate the rise and mass transfer for a single bubble, those results can be combined with an estimate of the bubbling rate (i.e., number of bubbles per time) to estimate the potential mass loading of methane to BML associated with dissolution from a steady release of bubbles. The estimate of the bubbling rate was based on the reported methane flux from the surface of BML [32], assuming that only bubble transport releases methane from the surface. The mass loading over a given depth interval is then given by the product of the bubble rate and the mass of the methane released from the bubble over that interval.

## 3. Results and Discussion

### 3.1. Temporal and Spatial Distributions of Water Column [CH_4(aq)_] and δ^13^C-CH_4(aq)_

Overall, the [CH_4(aq)_] decreased within BML over the course of this study. However, dynamic seasonal and annual trends emerged, particularly between winter ice-covered and ice-free periods. (Figure 3). The highest [CH_4(aq)_], where concentrations were more than 100 μM throughout the water column, were observed under the ice cover for both years that samples could be collected (2017, 2018). With the notable exception of spring 2017, upon the onset of ice-free conditions, the water column [CH_4(aq)_] decreased rapidly. The epilimnetic and metalimnetic concentrations decreased to, and remained at, less than 1 μM throughout the ice-free season. The [CH_4(aq)_] persisted in the hypolimnion, with the highest concentrations observed at the FWI early in the study (2015: 104 μM and 2016: 150 μM), subsequently decreasing to maximum concentrations of ~29 uM in 2017, and to ~27 uM in 2018. The [CH_4(aq)_] were always maximal at the FWI, decreasing away from the FFT up into the water column. The shallowest depth at which methane was present, i.e., the greatest penetration up into the water column from the FWI, also decreased from ~ 6 mbws at the bottom of the metalimnion in 2015 and 2016, to 7.5 mbws in 2017 (metalimnetic—hypolimnetic boundary) and 8.5 mbws (upper hypolimnion) in 2018. The exception to this trend was in the spring of 2017, when dissolved methane persisted in the epilimnion through May (~88 μM) and June (~25 μM), after which it decreased to <1 μM, consistent with other years (Figure 3). This unique behavior appeared to be related to the alum addition in the fall of 2016. Regardless of the persistence of methane in the spring of 2017, the overall trend was of a decreasing overall total [CH_4(aq)_] within the BML water column between 2015 and 2018 (Figure 3).

Concurrent δ^13^C-CH_4(aq)_ were only able to be determined for samples with sufficiently high [CH_4(aq)_] in the sample headspace for accurate detection, which was primarily samples in the hypolimnion in 2016, samples throughout the water column in spring 2017, and samples under ice in 2018 and 2019. The water column δ^13^C-CH_4(aq)_ in 2016 ranged from −64‰ to −58‰, with a mean of −62.2 ± 1.3‰. In February 2017, the upper water column (epilimnetic) δ^13^C-CH_4(aq)_ were consistently −60.9 ± 0.2‰ to a depth of 6.5 m, below which they became isotopically enriched as they approached the FWI, reaching a maximum of −57‰. In May and June of 2017, the δ^13^C-CH_4(aq)_ were generally isotopically enriched relative to February, ranging from −58‰ to −54.5‰. In February and March 2018, the δ^13^C-CH_4(aq)_ showed similar trends to February 2017. In February 2018, the δ^13^C-CH_4(aq)_ were again consistent in the epilimnion, with an average value of −61.7 ± 0.8‰ to a depth of 6.5 m, and below this point they became isotopically enriched to a value of −51.4‰, just above the FWI. In March 2018, the trend was similar, however, the δ^13^C-CH_4(aq)_ showed isotopic enrichment relative to the February data at all depths. The epilimnetic δ^13^C-CH_4(aq)_ had a mean value of −55.4 ± 1‰ to a depth of 5 m, below which they became isotopically enriched to a value of −42.4‰, just above the FWI.

### 3.2. Temporal and Spatial Distributions of FFT [CH_4(aq)_] and δ^13^C-CH_4(aq)_

Results from the 2016 and 2017 FFT sampling campaigns show that [CH_4(aq)_] increases sharply with depth, from <100 µM near the FWI, to >2000 µM within the upper 1.0 m of the FFT (Figure 4). From 1.0 to 3.5 m below the FWI, [CH_4(aq)_] values range from 2000 to 4000 µM and represent 60 to >80% of the theoretical CH_4(aq)_ saturation modeled using the measured temperature, pressure, and salinity values [33]. These thermodynamic models revealed that theoretical CH_4(aq)_ saturation increased approximately 3000 to 4000 µM with depth over the upper 4 m of FFT (Duan and Mao, 2006).

Concurrent δ^13^C-CH_4(aq)_ at the three platforms showed the greatest variability near the FFT surface, ranging from −74.8‰ to −61‰. Near-surface δ^13^C-CH_4(aq)_ were the most isotopically depleted at P3, and were the most enriched at P2. At depth, the δ^13^C-CH_4(aq)_ for all platforms were less variable, with a range of −69 to −63‰. The Δ^2^H-CH_4(aq)_ for these samples ranged from −289‰ to −326‰ (see Appendix A) 

### 3.3. Temporal and Spatial Trends in BML Microbial Community Abundance and Structure

Total water column PLFA concentrations, indicative of cellular abundances, ranged from 5 to 112 µg mL^−1^ over the course of the study (Figure 5), corresponding to estimated cellular abundances of 2 × 10^5^ to 5 × 10^6^ cells/mL on the basis of a conversion factor of 4 × 10^4^ pmol cell^−1^ [18]. These cellular abundances are similar to those reported in lakes ranging from Lake Erie [34] and Lake Michigan [35], to temperate lakes in Quebec [36], and to dystrophic lakes in Poland [37], as well as Lake Kinneret in Israel [38]. With the exception of the results for May 2017, the highest PLFA concentrations were consistently observed in the hypolimnion, consistent with the results of Risacher et al. (2018), and Arriaga et al. (2019), identifying the hypolimnion as the most microbiologically active zone in BML (Figure 5). Generally, the PLFA concentrations followed the same trend in both zones of the lake, with maximum concentrations observed in the earliest summer of the study, July 2015, and subsequently decreasing through 2016, and remaining consistent in 2017, with the notable exception of a spike in the PLFA concentrations observed in the epilimnion in May 2017. In 2018, there was an increase in the total PLFA in both zones in June, followed by a decrease in August 2018. 

Variations in the distributions of PLFA in BML also occurred, indicating that the changes in cellular abundances were contemporaneous with changes in the microbial community composition and/or metabolisms (Figure 5). Sixteen individual PLFA observed within BML were grouped according to their structures into: (i) Saturated PLFA produced by all microbes; (ii) C16:1 unsaturated PLFA (C16:1) that include biomarker PLFA for Type I and X methanotrophs [20]; (iii) C18:1 unsaturated PLFA (C18:1) that include biomarker PLFA for Type II methanotrophs [20] and are also produced by autotrophs [39,40]; (iv) Polyunsaturated PLFA (PUFA) that are produced by phototrophs and algae [41]; (v) Branched PLFA that include PLFA considered indicative of heterotrophic organisms, including sulphate reducers [19,42]; and (vi) Cyclic PLFA (Figure 5). Overall, and particularly early in the study (2015, 2016), the PLFA distribution was dominated by C16:1 and saturated PLFA. While C16:1 are not uniquely produced by methanotrophs, these high abundances are consistent with Type I/X methanotrophs comprising a large proportion of the microbial community within BML. Synchronous with the spike in the PLFA abundances in the epilimnion in May 2017 there was the most notable shift in the PLFA distributions as the C18:1 PLFA became the dominant PLFA and the proportion of polyunsaturated PLFA increasing markedly in both the epilimnion and hypolimnion. While C18:1 PLFA are produced by a variety of organisms, they are often associated with phototrophic organisms [39,40]. Similarly, PUFA are generally associated with eukarya, including algae [39]. This increase in the proportions of these PLFA is consistent with the bloom of algae in May 2017 that was indicated by field observations, increased water column Chl-a, and shifts in the microbial community structure, as indicated by genetic analysis in response to the alum addition [43]. Subsequently, the proportions of the C18:1 and PUFA PLFA decreased in 2018, concurrent with the re-establishment of an increased proportion of branched PLFA, suggesting a shift to an increased presence of heterotrophic organisms. At this time, cyclic PLFA that were present prior to the alum addition, were also re-established, though to a lesser extent than the branched PLFA. The cause of this shift is not clear, but it may related to changing metabolisms or to nutritional or stress conditions [44]. C16:1 again became the most abundant PLFA in the hypolimnion in 2018, indicating a return to the dominance of methanotrophy. In contrast, saturated PLFA dominated in the epilimnion with C18:1, continuing to be the second most abundant PLFA group, followed closely by branched PLFA. The concentrations of C16:1 in the epilimnion were quite a bit below these groups for all of 2018, suggesting a decreased role of methanotrophic organisms and an increased role for phototrophy and heterotrophy.

### 3.4. PLFA Isotopic Evidence of the Role of Microbial Methane Oxidation

The isotopic depletion of the C16:1 PLFA, relative to the C18:1 and C16:0 PLFA (Figure 6), demonstrated that microbial methane oxidation occurred throughout the water column over the entirety of the study. This interpretation is based on the general patterns of isotopic depletion expected between microbial PLFA and their carbon sources. PLFA produced by heterotrophic bacteria in aerobic environments, such as the BML water column, are generally ~3‰ depleted, relative to their carbon source [45]. While stronger isotopic depletions are seen in PLFA produced by anaerobic organisms grown under certain conditions [46], such anaerobic conditions were only observed in the BML water column in August in 2018 [43]. Because of the strongly isotopically depleted nature of methane, the PLFA of methanotrophs have been shown to be uniquely and strongly depleted in δ^13^C, and this can be a unique identifier of methanotrophy [47]. When applying this PLFA approach in the environment it is important to remember that the δ^13^C of an individual, or of grouped PLFA, will be determined by a mass balance of the isotopic compositions of the fatty acids derived from all organisms that produce a particular PLFA. Thus, a ubiquitously produced PLFA, such as C16:0, reflects an isotopic mass balance of all the organisms present in the community; similarly, the isotopic composition of pooled PLFA represents a mass balance of all the organisms producing PLFA included with the pool. 

Despite the potential for multiple sources to contribute to both the individual and pooled PLFA analyzed, the δ^13^C PLFA data show clear evidence of microbial methane oxidation throughout the BML water column. The δ^13^C of the C16:1 PLFA (δ^13^C = −59 to −40‰), which include the biomarker PLFA for Type I/X methanotrophs, was consistently the most depleted throughout the study. The most depleted end of the range of the C16:1 PLFA δ^13^C values closely approached the isotopic composition of the FFT methane endmember (δ^13^C = −61‰). In addition to this signature, the δ^13^C C14:0 PLFA (δ^13^C = −55 to −40‰), also produced in high proportions by Type I/X methanotrophs, see Ref. [20], were strongly depleted in 2016, when they could be measured. Such isotopic depletion of C16:1 PLFA in the groundwater near oil sands tailings ponds has previously been attributed to the uptake of a δ^13^C-depleted substrate, such as methane [48]. 

In contrast, the of C18:1 PLFA reflected the presence of a distinct component of the microbial community that was not utilizing methane-derived carbon, indicating that type II methanotrophy is not occurring in this system. The C18:1 PLFA were the most isotopically enriched and the most isotopically consistent PLFA over the course of the study, with a mean δ^13^C of −33.0 ± 2.0‰ and a range of only −32 to −33.6‰ across all depths for all years, with the exception of 2018, where δ^13^C C18:1 PLFA became slightly depleted (δ^13^C = −37‰) in the epilimnion in June, and the hypolimnion in August. The δ^13^C of C18:1 PLFA of −33.0 ± 2.0‰ is consistent with the heterotrophic utilization of a carbon source with a δ^13^C of ~−30‰ [49], and, thus, represents a component of the microbial community utilizing dissolved organic carbon sources, such as petroleum hydrocarbons from the underlying FFT, which have elsewhere been shown to be ~−30‰ [50], and/or photosynthetically produced organic carbon.

The relative contribution of these two end-member microbial carbon sources/metabolisms to the microbial community can be evaluated using the δ^13^C of the C16:0 PLFA, which lie intermediate between these two groups for almost all samples. As noted, C16:0 PLFA are produced by all organisms and, thus, reflect a mass balance of the isotopic compositions of the PLFA making up the microbial community. Therefore, the extent to which the C16:0 PLFA are comparable to one or other of the endmembers reflects the relative contribution that endmember is making to the overall microbial community. The most strongly depleted δ^13^C of C16:0 PLFA indicated that methane oxidation was making the greatest contributions to the overall microbial carbon cycling in the hypolimnion in 2016, consistent with the observation by Arriaga et al. (2019) and Riacher et al. (2018) that methane oxidation was occurring in this zone. While the extent of the isotopic depletion of δ^13^C of C16:0 PLFA was smaller, it continued to be isotopically depleted with respect to δ^13^C C18:1 PLFA at almost all depths and timepoints throughout the study, echoing the isotopic depletion of the C16:1 PLFA, reinforcing that methane oxidation was an important and recognizable component of microbial carbon cycling. 

### 3.5. Extent of Microbial Methane Oxidation

The extent of the impact of methane oxidation in removing dissolved methane from the water column can be evaluated using the observed shifts in δ^13^C-CH_4(aq)_ for the timepoints where it could be measured. The isotopic compositions of methane within the FFT determined in 2016 were δ^13^C-CH_4(aq)_ = −75 to −61‰ (Figure 4), and δ^2^H-CH_4(aq)_ = −343 to −289‰ (Appendix A"), indicative of microbial methane production by fermentative pathways [14]. While it was not possible to assess the controls on the range of δ^13^C-CH_4(aq)_ within the FFT with the data available, if it is assumed that the observed range is representative, then isotopic enrichment above this range within the water column can be considered to indicate the occurrence of fractionation during microbial methane oxidation [15]. 

Where concentrations were sufficient for analysis, the water column δ^13^C-CH_4(aq)_ ranged from values equivalent to the most isotopically enriched FFT values (δ^13^C-CH_4(aq_ = −63 to −61‰), to values as enriched as −45‰. The most isotopically enriched water column δ^13^C-CH_4(aq)_ values were generally observed close to the FFT, particularly in winter 2018, though isotopic enrichments were observed throughout the water column between successive sampling events in winter 2018 and in spring 2017 (Figure 3). 

The extent to which microbial methane oxidation was influencing the observed trends in the BML water cap [CH_4(aq)_] can be assessed by applying a Rayleigh fractionation model to the δ^13^C-CH_4(aq)_. Using the isotopic shift in the δ^13^C-CH_4(aq)_ values between two sampling points, and assuming an enrichment factor (ɛ) of 30‰ as representative for aerobic methanotrophy [15], the modeled fraction of methane remaining (f_iso_) can be compared to the observed changes in the concentration (f_obs_) (Table 1a,b). Applying this approach to the February and March 2018 data yielded estimates of methane oxidation based on changes in the δ^13^C-CH_4(aq)_ that were within 15% of the observed differences in [CH_4(aq)_]. This agreement indicates that the observed changes in [CH_4(aq)_] were predominantly the result of microbial methanotrophy occurring under the ice, with a net rate of 1.0 ± 0.3 μM d^−1^.

The same approach enabled the extent of methane oxidation that had occurred at the FWI to be assessed. Assuming an initial δ^13^C-CH_4(aq)_ of −61%, on the basis of the most enriched value observed within the FFT (Figure 4, Appendix A), the Rayleigh model indicated that the fraction of initial methane remaining in the samples just above the FWI in February 2017, and in February and March 2018, was 0.7 and 0.52, respectively, equivalent to an oxidation of 30 to 48% of the initial methane concentrations. Thus, without the occurrence of microbial methanotrophy at the FWI, the [CH_4(aq)_] would have been 1.3 to 2 times higher than what was observed at these times. 

The persistence of methane throughout the water column in spring 2017 allowed the contribution of methane oxidation to be differentiated from the nonisotopically fractionating processes of methane loss, such as air-partitioning (Table 1b). Between February and May 2017, the model estimated a fraction of the methane remaining (f_iso_) of a 0.88 to 0.91 equivalent to a loss by microbial oxidation of 9 to 12%, far lower than the observed decrease in [CH_4(aq)_] of 18 to 48% (f_obs_ = 0.52 to 0.82; Table 1b). Similarly, between May and June 2017, the fractionation-based estimates for microbial oxidation (8 to 26%: f_iso_ = 0.92 to 0.74) exhibited even greater deviation from the observed methane loss (47 to 81%: f_obs_ = 0.19 to 0.53; Table 1b). The fact that methane oxidation can only account for 23 to 45% of the observed loss between February and May, and 15 to 37% of the observed loss between May and June implies an equally important, if not greater, role for nonfractionating processes, such as air-water partitioning, in methane loss from the epilimnion over this interval. Air-water partitioning losses would be expected to occur annually at spring thaw and, in conjunction with the release of methane bubbles trapped beneath the ice, are likely responsible for the maximum fluxes of methane from the surface of BML observed by the eddy covariance measurements [32].

In addition to identifying the role of air-water partitioning, the isotope-based assessment of methane oxidation also indicated that the rate of net methane oxidation between February and May (0.22 ± 0.02 μM/day) was nearly an order of magnitude lower than that calculated for May to June (1.1 ± 0.4 μM/day), the latter being consistent with the rate calculated for February to March 2018. This low rate of methane oxidation between February and May 2018 suggested a large decrease in the oxidation capacity that may not only be related to decreased water temperatures, but that may also potentially be related to a decrease in the pelagic methanotrophic microbial community after the alum addition in the fall of 2016 (Jessen et al., this issue). 

### 3.6. Assessment of Methane Source Contributions

The PLFA and methane isotopic data provide clear evidence of microbial methane oxidation throughout the BML water column over the course of the study. Thus, the observed dissolved methane concentrations were the net outcome of the balance between the methane inputs and oxidation and imply that methane inputs were, in fact, larger than indicated by the measured BML water-column dissolved methane concentration data. The complete dataset, including the FFT methane concentrations collected in 2016, enabled the potential inputs related to: (i) The advection of porewater during FFT compaction; (ii) Molecular diffusion; and iii) The dissolution from bubbles (Figure 1) to be assessed. 

The [CH_4(aq)_] profiles were consistent, with the underlying FFT being the dominant source of CH_4(aq)_ to the water column, indicating that the advection of FFT porewater and/or molecular diffusion were the dominant sources. This is consistent with the observations of Dompierre et al. (2017), which identified the advection of FFT porewater as the primary source of chloride to the BML water column. Using the 2016 FFT porewater [CH_4(aq)_], and the rate of FFT-settling, calculated by Dompierre et al. (2017), of 0.73 m a^−1^ (Dompierre et al. 2017), the mass loading of methane due to porewater advection was calculated to be the dominant source of CH_4(aq)_ at 4.6 × 10^6^ to 1.8 × 10^7^ moles a^−1^, (equivalent to 55 to 220 metric tons of carbon per year) (Table 2). By comparison, the mass loading of CH_4(aq)_ by molecular diffusion from the FFT porewater, calculated using Fick’s law, ranged from 5.0 × 10^5^ to 8.7 × 10^6^ mol a^−1^ (6 to 104 metric tons of carbon per year) (Table 2). Notably, the upper end of this estimated range overlapped with the lower end of the range calculated for FFT porewater advection, indicating that molecular diffusion was potentially also an important source of CH_4(aq)_ to the hypolimnion in 2016. Since porewater advection due to FFT compaction is expected to decrease over time [52], molecular diffusion may become the dominant mechanism transporting CH_4(aq)_ to the hypolimnion in the future.

While advection and diffusion were the dominant sources of CH_4(aq)_ to the water column, the ongoing occurrence of ebullition represents another potentially important source. Using the model of Leifer and Patro (2002), the mass loading of methane to the water column, due to dissolution from bubbles during ebullition, was estimated to be 1.8 × 10^5^ to 4.7 × 10^6^ mol a^−1^ (2.1 to 56 metric tons of carbon per year) (Table 2), extensively overlapping with the range for molecular diffusion. It must be noted, however, that an interpretation of this result requires a consideration of the recognized large uncertainties associated with the modeled bubble dissolution. A methane flux of 0.0150 to 0.0345 mol m^−2^ d^−1^ from the lake surface, determined by eddy covariance measurements at Platform 1 [32], was used to estimate the number of bubbles moving through the water column and, thus, the overall methane loading. This eddy covariance method integrated a relatively small area of the lake surface, and on-site field observations indicate that the ebullition rates vary temporally and spatially over the lake, which adds uncertainty to the eddy covariance fluxes and, therefore, to the bubbling rates. The model was also sensitive to the estimates of bubble size, which, based on on-site field observations, can also vary temporally and spatially over large ranges. Despite these limitations, the fact that the estimated dissolved methane mass loading, due to dissolution from bubbles, is on the same order of magnitude as molecular diffusion does indicate that the dissolution from bubbles is potentially an important methane source to the water column. 

An important attribute of the dissolution from bubbles is that, unlike inputs from molecular diffusion or porewater advection that occur at the FWI into the hypolimnion, the dissolution from bubbles occurs throughout the water column. Such a mechanism is consistent with the observation of evidence of methane oxidation within the epilimnion throughout the study, despite the low concentrations of dissolved methane and previous evidence indicating high rates of methane consumption in the hypolimnion [5,7] that would be expected to prevent the upward diffusive transport of methane across the thermocline. The dissolution of methane from bubbles also provides an explanation for the observation of high concentrations of dissolved methane in the epilimnion during under-ice sampling. In February of both 2017 and 2018, relatively consistent concentrations of methane, with invariant δ^13^C values of −61‰, equivalent to the enriched end of the FFT range, were observed across the epilimnion. This is consistent with the delivery of methane to the epilimnion without experiencing the isotopic fractionation occurring in the hypolimnion near the FWI observed at the same time. This would be expected for bubbles that are released from the FFT and trapped beneath the ice, where they dissolve into the epilimnion waters and are then circulated through the epilimnion. While the dissolved methane appears to have undergone little fractionation in February, there is clear evidence of isotopic fractionation between February and March 2018 throughout the water column. The reason for the lack of evidence for the under-ice methane oxidation in February, and the subsequent establishment of an oxidized signal in March 2018, is not known. However, it may be related to the relative rates of methane oxidation versus supply, suggesting that the under-ice oxidation prior to March is relatively slow. 

### 3.7. Biogeochemical Response to Perturbation, Spring 2017: Phototrophic Response to Alum Addition

As noted, the low rates of methane oxidation and the persistence of CH_4(aq)_ into May and June in 2017 (Table 1b) appear to be the result of the perturbation of the BML system by the whole-lake addition of alum in fall 2016. The decreased rates of methane oxidation observed between February and May 2017 are consistent with a decreased presence of methanotrophic organisms within the water column, which would be expected if the alum addition had stripped out pelagic organisms as well as particulates from the water column. Because of technical issues during analysis, the PLFA abundances and distributions could not be confidently determined for February 2017. However, the PLFA abundances (Figure 5) and isotopic compositions (Figure 6) indicate that the pelagic microbial community underwent a significant shift immediately following this interval. The total PLFA abundances in the epilimnion increased dramatically in May 2017 (Figure 5), and there was a concurrent shift to a greater proportion of the biomarkers of phototrophic organisms (C18:1unsat and polyunsaturated PLFA) in both the epilimnion and hypolimnion (Figure 5). These trends are consistent with the occurrence of a bloom of phototrophic organisms, such as algae, indicated by field observations, increased water column Chl-a, and shifts in the microbial community structure [43], associated with increased water clarity after the alum addition in fall 2016. The δ^13^C of the PLFA further support this interpretation. In May 2017, the δ^13^C of all the epilimnetic PLFA became isotopically enriched (Figure 6). This was one of only two times that the δ^13^C of the C16:1 PLFA was >−40‰ and was only slightly depleted relative to the C16:0 and C18:1 PLFA. The relatively small isotopic depletion of the C16:1 relative to the other PLFA indicates that while methane oxidation was ongoing, it represented a relatively smaller component of the overall microbial community than at other times. Furthermore, the relatively close agreement between the δ^13^C of the C16:0 and C18:1 PLFA was consistent, with these PLFA being derived from a similar photosynthetically derived carbon source. Consistent with the re-establishment of higher rates of microbial methane oxidation between May and June 2017, the C16:1 PLFA again became strongly isotopically depleted throughout the water column, although the relatively small isotopic depletion of C16:0 PLFA indicates that this process was making a relatively smaller contribution to the overall microbial community than previously, in 2016. 

### 3.8. δ^13^C Evidence of Ongoing Shifts in Microbial Metabolism in 2018

Methanotrophy continued to play a recognizable role in the hypolimnion in June and August 2018, and in the epilimnion in June 2018, where the δ^13^C of C16:1 PLFA remained isotopically depleted, echoed by the isotopic depletion of the C16:0 PLFA (Figure 6). In contrast, there was a large shift in the δ^13^C of the PLFA in the epilimnion in August, such that the δ^13^C of the PLFA groups converged to relatively enriched values of −29 to −34‰. This shift is consistent with what was observed in May 2017 and suggests a shift to a dominance of phototrophic metabolism. Increased phototrophy in August would be consistent with the general trend of increasing water clarity over the summer [23]. While the signal of phototrophy, based on the C18:1_unsat_ and polyunsaturated PLFA abundances in August 2018, was not as strong as that observed in May 2017, both of these PLFA groups continue to make large contributions to the PLFA distribution, which is dominated by ubiquitously produced saturated PLFA (Figure 5). 

A second change in the microbial metabolism, observed in the summer of 2018, was indicated by the shift in the δ^13^C of C18:1 PLFA to −37‰ in the epilimnion in June, and in the hypolimnion in August (Figure 6). While not as large a shift as observed for C16:1 PLFA associated with methanotrophy, it is notable given the consistency of all of the previous samples, which ranged from −32 to −34‰. The cause of this shift is not certain; however, it is likely the outcome of a change in the metabolisms within the BML microbial community [43]. Jessen et al., 2021, observed that the bottom waters of BML became anoxic in August 2018, and further detected the presence of dissolved aqueous sulphide in the water column for the first time. Microbial sulphate reduction is known to be able to produce isotopically depleted PLFA [46]. An increased presence of heterotrophic bacteria, including sulphate-reducing bacteria, would also provide an explanation for the increased abundances of branched PLFA that are often produced in abundance by these organisms. While the current data are insufficient to definitively assess the cause of the observed shift in the isotopic compositions, the unique δ^13^C PLFA, observed in both the epilimnion and hypolimnion in August 2018, indicate ongoing changes in the dominant microbial metabolisms in the lake that are consistent with the shifts in the microbial community composition reported in Jessen et al. (submitted) [43]. 

## 4. Conclusions

The results of this study demonstrate that microbial methane oxidation made an important contribution to limiting methane release throughout the BML water column over the entirety of the study (2015–2019). These results confirm previous reports indicating the occurrence of microbial methane oxidation, and demonstrate that this process responded dynamically to both seasonal cycles and management intervention. The PLFA abundances, distributions, and isotopic compositions provided clear evidence of extensive methane oxidation, which was greatest in the first year of the study, particularly in the hypolimnion, but remained recognizable throughout the water column at all sampling times. The dissolved methane concentration profiles and the modeling results indicate that the porewater advection and molecular diffusion from the FFT were the primary sources of methane to the BML water column. The observed isotopic fractionation of methane close to the FWI demonstrated that the methane concentrations had undergone oxidation, and inputs from the FWI were 1.3 to 2 times higher than the measured concentrations suggested. The mass balance modeling results also indicate that methane dissolution from bubbles was an important source of dissolved methane that could account for the observations of detectable dissolved methane within the epilimnion throughout the study, as well as for the observed under-ice epilimnetic high concentrations of dissolved methane in 2017 and 2018. While there was no evidence of methane fractionation in the February (under ice) data for both 2017 and 2018, shifts in δ^13^C CH_4_ in March 2018 indicated that the occurrence of methane oxidation was responsible for the observed decreases in the methane concentrations throughout the water column. Taken together, these results indicate that microbial methane oxidation is capable of preventing the release of dissolved methane from BML. Thus, if methane ebullition releases are reduced over time, or because of mitigation efforts, the BML water column should prevent BML from being a long-term source of methane to the atmosphere. 

The addition of alum to the lake in the fall of 2016 had extensive impacts on the microbial biogeochemical cycling within BML. The calculated rates of methane oxidation between February and May 2017 were nearly an order of magnitude lower than those at other times. This decreased rate was associated with the persistence of dissolved methane throughout the water column of BML into June 2017. The decreased rates of methane oxidation during this time may reflect the stripping out of pelagic methanotrophs by the alum addition. The PLFA and methane δ^13^C results indicate that the methane oxidation by the pelagic microbial community had re-established in May 2017, and even more strongly in June (2017), though this process was a relatively smaller contribution to the overall microbial community. Shifts in the PLFA distribution and δ^13^C indicated the occurrence of a phototrophic algal bloom that was further supported by field observations. These observations were consistent with the trends observed by Jessen et al. [43]. The observation of such a shift, and the associated change in the lake's capacity to oxidize methane, demonstrates that, in addition to having the planned effect of increasing the water clarity, the alum addition appears to have had large impacts on the biogeochemical cycling within the lake that should be considered when undertaking management interventions.

The biogeochemical cycling within BML continued to change in 2018. While methane oxidation continued to play a recognizable role, the PLFA distributions and isotopic compositions indicated the emergence of a phototrophically dominated community in the epilimnion in August 2018, similar to that observed in May 2017. However, unlike in May 2017, the influence of this community shift is smaller in the hypolimnion, where methanotrophy still plays a larger role. Concurrent with this is the observation of a shift in the carbon cycling by the other components of the microbial community, indicated by a shift in the δ^13^C of C18:1 PLFA. While the specific causes of these shifts are not yet clear, these ongoing changes in the PLFA and isotopic biomarkers of the BML microbial community are concurrent with the changes identified via amplicon sequencing by Jessen et al. [43]. 

These results demonstrate the dynamic nature of microbial methane oxidation as a carbon and energy source to the BML microbial community, and as a control on the methane and oxygen concentrations. The role of methane oxidation was dramatically affected by the addition of alum in the fall of 2016 to increase the water clarity, such that methane persisted into June 2017, for the first and only time observed in the study period (2015–2018). The increased water clarity led to a phytoplanktonic bloom in May 2017 that continued to influence the BML microbial community throughout 2017. Further shifts in the microbial methane oxidation and carbon sources were indicated in 2018. Whether these were influenced by the 2016 alum addition or represent new changes in the lake as methane sources and other biogeochemical parameters continue to develop, is unclear. As noted, on the basis of these results, it appears that microbial methane oxidation will continue to be capable of consuming the dissolved methane inputs to BML, particularly as the methane inputs are expected to decline. However, the ongoing changes in the microbial community in 2018 indicate that other microbial metabolic processes may become dominant within the water column. If these changes lead to major shifts in the redox conditions and/or microbial metabolisms within the lake, they may impact the biogeochemical balance and affect the extent of methane oxidation, which could have important implications for lake management and which needs to be the focus of further investigations. 

## Figures and Tables

**Figure 1 microorganisms-09-02509-f001:**
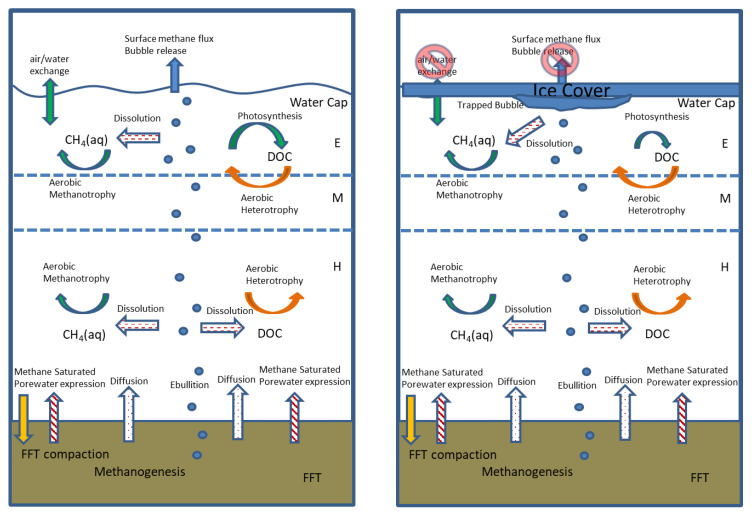
Schematic showing the potential pathways of microbial methane transport and consumption within BML under ice-free and ice-covered conditions. The primary distinctions under ice-covered conditions are the lack of loss due to bubble release, and/or the air water partitioning and potential slowing of rates of microbial methanotrophy and, instead, the buildup of bubbles and dissolved methane under the ice. E, epilimnion; M, metalimnion; H, hypolimnion.

**Figure 2 microorganisms-09-02509-f002:**
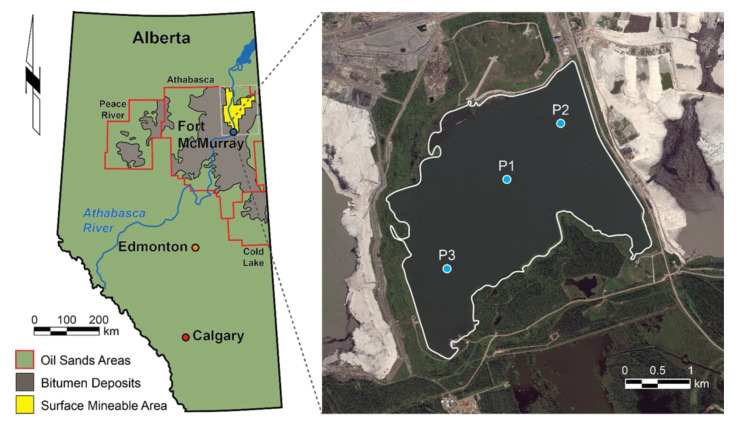
Sampling site showing the distribution of oil sands in northern Alberta and the location of Base Mine Lake (BML: black dot). The second panel shows a close up of BML showing the locations of the three primary sampling platforms referred to in this study.

**Figure 3 microorganisms-09-02509-f003:**
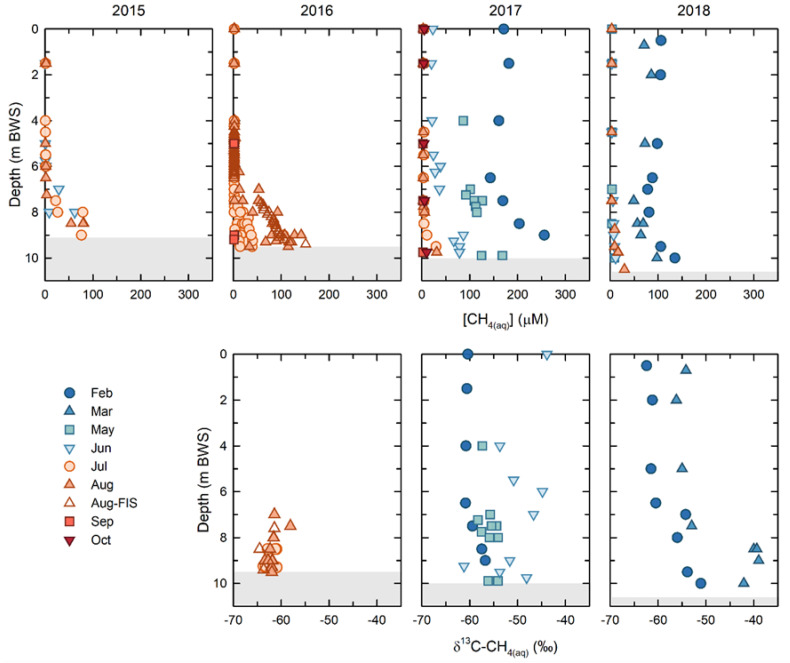
Dissolved methane concentrations (μM) and isotopic compositions (δ^13^C) for 2015, 2016, 2017, and 2018. Under-ice samples (February and March) were only obtained in 2017 and 2018. The shaded region at the bottom shows the depth of the FFT and its settlement over time.

**Figure 4 microorganisms-09-02509-f004:**
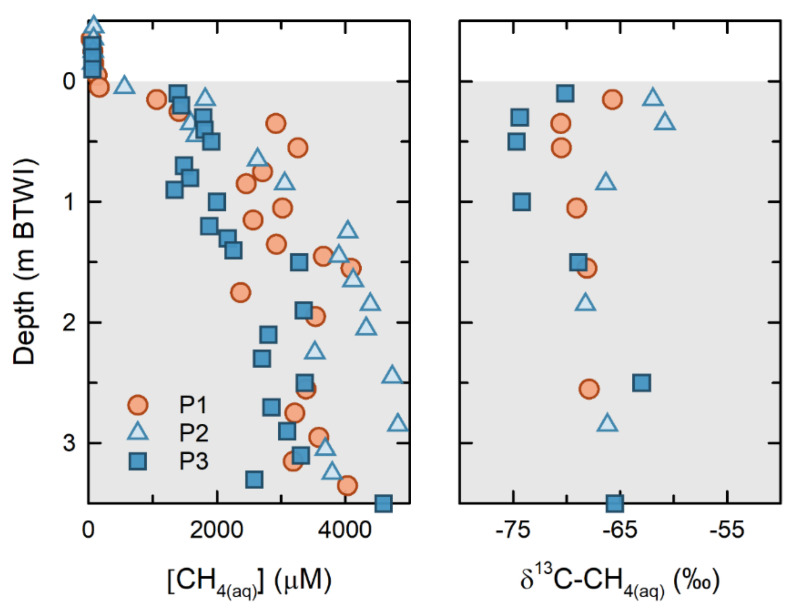
Dissolved methane concentrations (μM) and isotopic compositions (δ^13^C) for FFT porewater in 2016 from P1, P2, and P3 (see sampling site locations on Figure 2).

**Figure 5 microorganisms-09-02509-f005:**
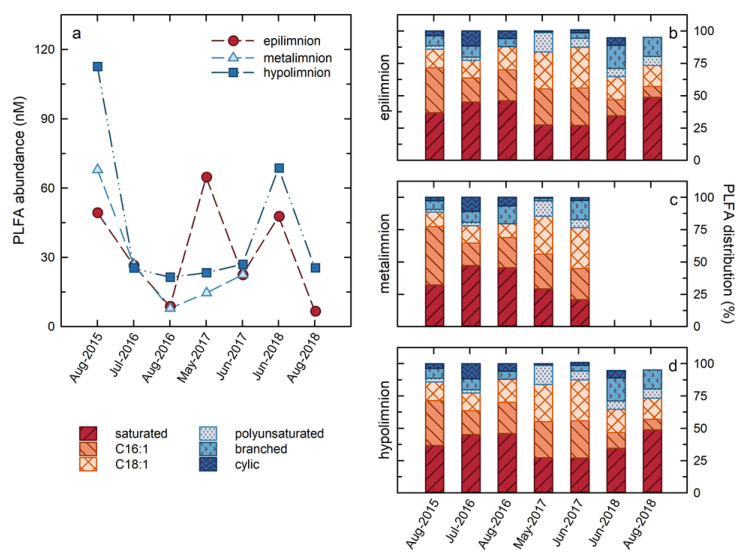
Total PLFA concentrations for the epilimnion, metalimnion, and hypolimnion over the course of the study (panel (**a**)). Distribution of PLFA groups in the epilimnion (**b**), metalimnion (**c**), and hypolimnion (**d**) over the course of the study.

**Figure 6 microorganisms-09-02509-f006:**
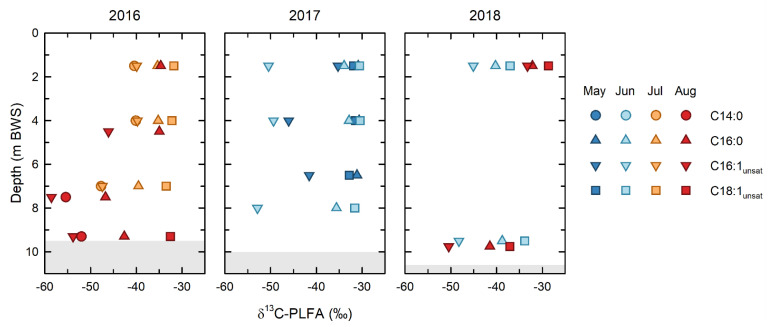
PLFA isotopic compositions for indicator PLFA, including pooled C16:1 PLFA, which include methanotroph biomarker PLFA, C14:0 PLFA also produced by methanotrophs (2016 only), pooled C18:1 PLFA produced by non-methanotrophic organisms, and C16:0 produced by all organisms. Depth is below water surface.

**Table 1 microorganisms-09-02509-t001:** Methane loss modeling. Comparison of calculated and observed extents and rates of methane. Observed extent of reaction presented as fraction of reactant remaining (fobs) determined by comparison of measured methane concentrations at comparable depths. Calculated extent of reactions based on extent of isotopic fractionation of δ^13^C CH_4_ for comparable depth, and calculated using the Rayleigh fractionation model, εlnf = ln(((δ13Cr/1000) + 1)/((δ13Cro/1000) +1)), as per Slater 2003 [51], assuming an isotopic enrichment factor of −30‰ [15]. Table 1 reports assessment of under-ice methane oxidation between February and March 2018. In 2018, f_obs_ and f_iso_ were within 15%, and often within 5%. Linear rate of methane loss calculated from f_iso_ data for consistency to 2017 (Table 1b). Table 1 reports results for intervals between under-ice conditions and ice off (Feb–May), and during spring (May to June) in 2017.

(a) 2018														
Depth (m)	CH4 (μM)			δ13C CH4 (‰)			f (obs)		f (iso)		linear Rate			
	Feb	Mar		Feb	Mar						(uM/day)			
0.5	103	68		−62.7	−54.5		0.66		0.76		1.2			
2	102	83		−61.5	−56.5		0.81		0.85		0.6			
5	95	69		−61.8	−55.3		0.73		0.8		0.9			
7	75	46					0.61				1			
8	78	54		−56.3	−40.4		0.69		0.59		0.8			
9	102	61		−54.2	39.3		0.6		0.61		1.4			
10	132	95		−51.4	−42.4		0.72		0.74		1.2			
									Mean		1			
									stdev		0.3			
**(b) 2017**														
	**CH4 (μM)**			**δ13C CH4 (‰)**			**f (obs)**		**f (iso)**		**Linear Rate (uM/day)**		**Linear Rate (iso)**	
**Depth (m)**	**Feb**	**May**	**June**	**Feb**	**May**	**June**	**Feb–May**	**May–June**	**Feb–May**	**May–June**	**Feb–May**	**May–June**	**Feb–May**	**May–June**
0	169	87	21	−61		−44	0.52	0.24			0.9	2.5		1.36
1.5	179	93	18	−61			0.52	0.19			1.0	2.8		
4	159	84	19	−61	−58	−54	0.53	0.22	0.89	0.89	0.8	2.4	0.19	0.66
6.5	141	115	34	−61			0.82	0.30			0.3	3.0		
7.5	167	124	36	−60	−56	−47	0.74	0.29	0.88	0.74	0.5	3.2	0.21	1.59
8.5	201	113	60	−58	−54		0.56	0.53	0.89		1.0	1.9	0.24	
9	253	166	77	−57	−54	−52	0.66	0.46	0.91	0.92	1.0	3.3	0.24	0.97
										Mean	0.8	2.7	0.22	1.14
										stdev	0.3	0.5	0.02	0.41

**Table 2 microorganisms-09-02509-t002:** Methane source mass loading estimates. Calculated inputs of methane for advection, molecular diffusion, and dissolution from bubbles.

Process	mol CH_4_/yr		Metric Tons/yr	
	Max	Min	Max	Min
Advection	1.8 × 10^7^	4.6 × 10^6^	220	55
Diffusion	8.7 × 10^6^	5.0 × 10^5^	104	6.1
Bubble dissolution	5 × 10^6^	2 × 10^5^	56	2.2

## Data Availability

Not applicable.

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
