# Peer review of "Isotopic and Chemical Assessment of the Dynamics of Methane Sources and Microbial Cycling during Early Development of an Oil Sands Pit Lake"

_microorganisms, 2021, doi:10.3390/microorganisms9122509_

Round 1

Reviewer 1 Report

Microorganisms Review 11-2021

The reviewer compliments the authors on an interesting manuscript involving this reclamation strategy for tailings and the details of the four-year study from a chemical and microbiological prospective.  Overall, the paper is scientifically sound and well-written. The reviewer has comments for consideration and questions that should be addressed. There are areas where there is confusion with table labels and where they are identified in the text that need clarification.  The supplementary material requires minor adjustment.

Introduction:

The reviewer likes the introduction but realized that other readers may not be familiar with methanotrophy, stable isotopes and PLFA methods. It might be suggested that a small paragraph or two be added to address these issues. With methane oxidation and membrane lipids, the authors mention type, I, II and X methanotrophs later in the manuscript, but fail to mention ammonia oxidizer that have multiple membrane systems that can also oxidize methane.  [These organisms might not be important in this system; however, the reviewer is interesting on the complimentary paper by Jessen as it might have provided light on this issue as well as the potential to discrimination between type I and II methanotrophs (which is more fitting later in manuscript)].

Figure 1. The letters, E, M, and H should be defined in the legend.

Materials and Methods:

Lines 143-144 has an interesting font change that should be corrected.

PLFA methods- The reviewer understands the issues associated with field sampling and preservation and assumes the filtering could not be done in the field.  There are some concerns that the freeze thawing may have affected some cells causing lysis that was then lost upon filtration.  It is understandable that the authors filtered to differentiate between attached and pelagic microbes, rather than freeze drying the entire liter and analyzing the total community. 

The reviewer is interesting in the PLFA measurements for the glass filters (if analyzed) which should have contained attached communities as well as some eukaryotes. This information is not necessary for this manuscript.

Lines 254-255 requires clarification as the method generally uses methanolic KOH and that the methanol (for both the methanolic KOH and the Toluene:methanol solution) used in the methylation protocol was isotopically characterized.

Was there enough PLFA recovered (between regular FAME analysis and stable isotope analysis) to have preform a dimethyl disulfide derivation?  If so, those results would have added support to the type I vs type II methanotrophy discussion by showing the presence of the 16:1w8 or 18:1w8 biomarkers which could be used to define methanotrophy vs heterotrophic abundances.

Results and Discussion:

Figure 5. need work.  Panels a and b are not labeled.  The saturated fames have a poly associated with it that does not belong.  The legend is missing metalimnion in the second sentence.  The text from line 397 to 403 might be best stated in the legend.

Lines 394 to 424.  In this section there is no discussion of the cyclic FAMEs.  It is interesting that the cyclic FAMEs are higher in samples before and much later after alum addition.  Is this more an issue of detectable PLFA biomass or does it relate to nutritional or stress conditions in the BML. The branched and cyclic FAMEs could potentially increase with anoxic conditions. Did the types and abundance of branched FAMEs change over time (thus showing community changes).

 This section would benefit from the DMDS analysis, but it is not necessary for the manuscript.

There are mixed messages regarding specific PLFA that should be clarified a bit more.  Two examples- saturated PLFA produced by all microbes, yet later C14:0 is attributed to type 1 methanotroph; C18:1s. type II methanotrophs, heterotrophs and precursors for eukaryotes.  With respect to the second example the authors try to explain it as non-methanotrophic organisms which the reviewer believes is a reasonable way to explain it.  However, there could be text regarding the lack of detection of type II methanotrophs (Are these more attached microbes that the type I?). Possibly a bit more text is required here. 

Tables 1a, 1b have no titles or legends

All tables identified in text need to be check for proper citation.

Lines 458 section – not utilizing methane derived carbon, therefore type II methanotroph is not observed.

Comment: Addressing anaerobic methane oxidation in anoxic waters does not appear to be required in this manuscript.

Conclusion:

The overall changes in microbial community and process is properly described based on the data provided.

In introduction at the end of the first paragraph, the authors discuss reclamation strategy yet in the conclusion this theme is not addressed and should be at least speculated about briefly in the conclusion(other than to say more studies are needed).

Comment for consideration:

The PLFA method does not allow for detection of Archaea. – should this be mentioned in the text, probably not because the study appears to an aerobic system. It is known that in bulk aerobic environments anaerobic process can still be active; however, it is more associated with particulate matter.  Yes, more investigations are required. 

Supplementary Material:

Just below S.8, there is text regarding parameter listings.   The reviewer does not feel a listing is necessary, but other reviewers may suggest a table be added to the supplementary document.

Author Response

The reviewer compliments the authors on an interesting manuscript involving this reclamation strategy for tailings and the details of the four-year study from a chemical and microbiological prospective.  Overall, the paper is scientifically sound and well-written. The reviewer has comments for consideration and questions that should be addressed. There are areas where there is confusion with table labels and where they are identified in the text that need clarification.  The supplementary material requires minor adjustment.

 Response 1: Thank you to the reviewer for the positive feedback.  We have edited the manuscript according to the reviewers comments as described below.

Introduction:

The reviewer likes the introduction but realized that other readers may not be familiar with methanotrophy, stable isotopes and PLFA methods. It might be suggested that a small paragraph or two be added to address these issues. With methane oxidation and membrane lipids, the authors mention type, I, II and X methanotrophs later in the manuscript, but fail to mention ammonia oxidizer that have multiple membrane systems that can also oxidize methane.  [These organisms might not be important in this system; however, the reviewer is interesting on the complimentary paper by Jessen as it might have provided light on this issue as well as the potential to discrimination between type I and II methanotrophs (which is more fitting later in manuscript)].

 Response 2: Two paragraphs have been added to the introduction to provide a brief overview of the application of stable isotopic analysis and PLFA analysis. 

Figure 1. The letters, E, M, and H should be defined in the legend.

 Response 3: This change has been made

Materials and Methods:

Lines 143-144 has an interesting font change that should be corrected.

 Response 4: The issue with font changes has been addressed

PLFA methods- The reviewer understands the issues associated with field sampling and preservation and assumes the filtering could not be done in the field.  There are some concerns that the freeze thawing may have affected some cells causing lysis that was then lost upon filtration.  It is understandable that the authors filtered to differentiate between attached and pelagic microbes, rather than freeze drying the entire liter and analyzing the total community. 

The reviewer is interesting in the PLFA measurements for the glass filters (if analyzed) which should have contained attached communities as well as some eukaryotes. This information is not necessary for this manuscript.

 Response 5: We agree with the reviewer that the filters could contain eukaryotic organisms and have identified in the new paragraphs in the introduction that eukaryotes are captured by PLFA analysis, the presence of eukarya within the PLFA was also directly identified within the paragraph describing the PLFA distributions.

Lines 254-255 requires clarification as the method generally uses methanolic KOH and that the methanol (for both the methanolic KOH and the Toluene:methanol solution) used in the methylation protocol was isotopically characterized.

 Response 6: This has been clarified

Was there enough PLFA recovered (between regular FAME analysis and stable isotope analysis) to have preform a dimethyl disulfide derivation?  If so, those results would have added support to the type I vs type II methanotrophy discussion by showing the presence of the 16:1w8 or 18:1w8 biomarkers which could be used to define methanotrophy vs heterotrophic abundances.

 Response 7: There were not sufficient PLFA to do DMDS for the water samples.  However, as noted by the reviewer, and re inserted in this manuscript around line 458 (see below), the isotopic data does not support type II methanotrophs.

Results and Discussion:

Figure 5. need work.  Panels a and b are not labeled.  The saturated fames have a poly associated with it that does not belong.  The legend is missing metalimnion in the second sentence.  The text from line 397 to 403 might be best stated in the legend.

 Response 8: This figure had been corrected and metabolimnion has been added to the legend. We did not move the text to the legend to ensure that the reader was clear about the groupings when reading the text.

Lines 394 to 424.  In this section there is no discussion of the cyclic FAMEs.  It is interesting that the cyclic FAMEs are higher in samples before and much later after alum addition.  Is this more an issue of detectable PLFA biomass or does it relate to nutritional or stress conditions in the BML. The branched and cyclic FAMEs could potentially increase with anoxic conditions. Did the types and abundance of branched FAMEs change over time (thus showing community changes).

 Response 9: We have edited this section to clarity that the branched and cyclic PLFA were re-established after the impacts of alum addition.  We have noted that the cyclic PLFA may indicate changes in nutrient or stress state or that both may relate to changes in microbial metabolisms.  But we do not feel there is sufficient information to be definitive about the causes of the observed trends without further data.

 This section would benefit from the DMDS analysis, but it is not necessary for the manuscript.

 Response 10: We agree that further characterization would have elucidated more of the PLFA present.  However, it was not achieved and as the reviewer notes, was note required for this manuscript. 

There are mixed messages regarding specific PLFA that should be clarified a bit more.  Two examples- saturated PLFA produced by all microbes, yet later C14:0 is attributed to type 1 methanotroph; C18:1s. type II methanotrophs, heterotrophs and precursors for eukaryotes.  With respect to the second example the authors try to explain it as non-methanotrophic organisms which the reviewer believes is a reasonable way to explain it.  However, there could be text regarding the lack of detection of type II methanotrophs (Are these more attached microbes that the type I?). Possibly a bit more text is required here. 

 Response 11: We agree and have re-inserted the comment that the isotopic results for the C18:1 PLFA indicate that Type II methanotrophy is not a dominant process.  With respect to the C14:0 we had noted that they are produced in high proportions by Type I/X methanotrophs within the sentence.  We agree, they are not unique to these organisms, similarly to the fact that C16:1 are not unique to methanotrophs.  However, we do feel that the similarity in isotopic composition of these PLFA supports the presence and metabolism of the type I/X methanotrophs.  We have added a reference to the statement about the C14:0 PLFA to increase the clarity.

Tables 1a, 1b have no titles or legends

 Response 12: The table legend was incorrectly formatted and appeared as text.  It has been reformatted.

All tables identified in text need to be check for proper citation.

Response 13: This change has been made 

Lines 458 section – not utilizing methane derived carbon, therefore type II methanotroph is not observed.

 Response 14: This change has been made

Comment: Addressing anaerobic methane oxidation in anoxic waters does not appear to be required in this manuscript.

 Response 15: We agree, certainly our ongoing work to investigate methane cycling within the FFT will focus on this potential process and its role.

Conclusion:

The overall changes in microbial community and process is properly described based on the data provided.

 In introduction at the end of the first paragraph, the authors discuss reclamation strategy yet in the conclusion this theme is not addressed and should be at least speculated about briefly in the conclusion(other than to say more studies are needed).

 Response 16: Further comments have been added to the conclusions discussing the implications of these results to lake management.

Comment for consideration:

The PLFA method does not allow for detection of Archaea. – should this be mentioned in the text, probably not because the study appears to an aerobic system. It is known that in bulk aerobic environments anaerobic process can still be active; however, it is more associated with particulate matter.  Yes, more investigations are required. 

 Response 17: A comment clarifying that PLFA does not measure Archaea but does combine both bacteria and eukarya was added to the introduction to clarify this point.  Our further work will investigate archaeal ether lipids as well.

Supplementary Material:

Just below S.8, there is text regarding parameter listings.   The reviewer does not feel a listing is necessary, but other reviewers may suggest a table be added to the supplementary document.

 Response 18: Thank you to the reviewer for catching this holdover.  The phrase has been removed.

Reviewer 2 Report

Here, the authors utilise a variety of geochemical and isotopic (δ13C) analyses to understand microbial methane cycling in an Athabasca oil sands region (AOSR) pit lake (Base Mine Lake) over a four-year period from 2015 to 2018. Distributions and δ13C values of phospholipid fatty acids (PLFAs) demonstrated that dissolved methane originating from underlying fluid fine tailings (FFT) was an important carbon source for the water column microbial community at all sampling times. Mass balance modelling results indicated that methane dissolution from bubbles was an important source of dissolved methane. Furthermore, the addition of alum to the lake in fall of 2016 had major impacts on microbial biogeochemistry.

Overall, this is a well-written manuscript that makes a significant contribution to our understanding of biogeochemical processes in oil sands pit lakes, which are considered to be one of the leading aquatic reclamation strategies in the AOSR. The discussion is supported by the data presented and the arguments are generally easy to follow. There are however several parts of the manuscript that require additional information and rearrangement of text. These instances and additional comments and suggestions are provided below:

Abstract, line 23:

Define FFT

Line 24:

The second ‘throughout the water column’ is redundant and can be removed.

Line 52:

Concentrations of PAHs and alk-PAHs in OSPW are generally not very high. Naphthenic acids are typically the main organic contaminant of concern in tailings ponds.

Figure 1:

On the right-side figure, I suggest making the crossed-out circles more transparent so that the underlying text can be seen.

Figure 1 heading (lines 78-79):

What about the role of temperature in controlling the rates of methanotrophy / heterotrophy? Wouldn’t the rates be lower in winter due to lower water temperatures?

Line 178-180:

I find the description of parallel samples confusing. If 30 mL of water was used for CH4 concentrations, and 50 mL of water was used for isotopic analyses, what was the 60 mL of water that was collected used for?

Line 191:

What was the volume of the headspace? 60 mL? Did the headspace volume vary considerably?

Line 206:

How much helium did you know to add to reach atmospheric pressure?

Line 213:

How did you calculate aqueous CH4 concentrations from headspace concentrations? i.e., what proportion of CH4 did you assume was distributed into the headspace?

Line 247:

Were the PVDF filters cleaned prior to use?

Line 255:

I assume the authors meant to say ‘isotopically characterized methanolic KOH’. Also, I suggest moving ‘δ13C’ to before KOH i.e., ‘isotopically characterized (δ13C) methanolic KOH’.

Also, I assume that the methanol in the 1:1 toluene:methanol mixture was isotopically characterised?

Line 277:

Should be ‘isotopically characterized methanolic KOH…”

Line 296:

What water temperatures did this model assume?

Line 359:

Should be Figure 3.

Somewhere below line 359:

Considering the data are shown in Figure 3, this section is missing a discussion on general trends in δ13C-CH4 values in the water column.

Line 364:

I suggest adding the following sub-section title: ‘Fluid Fine Tailings (FFT)’, or something similar.

As for the missing discussion on water column δ13C-CH4 values, this section needs a discussion on overall trends in δ13C-CH4 values from FFT.

Figure 4 heading (line 374):

Add the following to the end of the text: (see sampling site locations on Figure 2).

Line 398-399 (and throughout manuscript):

The terms 16:1 and 18:1 already indicate an unsaturated PLFA; thus the use of the suffix ‘unsat’ is redundant and can be removed.

Line 399:

For an AOSR-relevant study demonstrating significant 13C-depletion in 16:1 PLFAs as a result of methanotrophy (similar to that reported by the authors here), see Ahad et al., 2018, Science of the Total Environment.

Lines 411-412:

“While C18:1unsat PLFA are produced by a variety of organisms, they are often associated with phototrophic organisms.” References?

Line 431:

The 4-6‰ depletion in PLFAs compared to carbon source for heterotrophic bacteria in aerobic environments stated by the authors is a bit higher than that reported in the literature. In general, this difference is closer to 3‰. For instance, the paper cited by the authors (Hayes, 2001) states the following (page 23):

“A third approach was taken by Blair et al. (1985) who grew E. coli on glucose while obtaining a complete mass balance. They confirmed the finding of Monson and Hayes (1982a) that the fatty acids are depleted in 13C by 3‰ relative to the glucose carbon supply.”

Other investigations have also reported an approximate 3‰ rather than up to 6‰ depletion in PLFAs relative to carbon source (e.g., Boschker et al., 2005, Limnol & Oceanogr; Teece et al., 1999, Org. Geochem.), including one relevant study that examined microbial carbon sources in surface sediments from AOSR tailings ponds (Ahad et al., 2013, Environ. Sci. Technol.).

Line 441:

What exactly do the authors mean when they refer to the isotopic compositions of ‘pooled’ PLFAs? Do the authors mean to say that several peaks were integrated together using GC-IRMS software?

Line 464:

See comment above for line 431. AOSR bitumen has a δ13C value of around -30‰ (Ahad et al., 2013, Environ. Sci. Technol.); thus microbial utilisation of residual bitumen-derived hydrocarbons in FFT would be consistent with δ13C values close to -33‰ in PLFAs.

Line 500:

Should be Figure 3.

Line 528:

What is the value of -61‰ based on? The most 13C-enriched δ13C-CH4 value found in FFT?

Lines 555-558:

Could lower rates of methane oxidation during Feb-May 2018 have also been caused by lower water temperatures?

Line 648:

Should be -40‰.

Lines 682-685:

This sentence is awkward and needs to be reworded.

Author Response

Here, the authors utilise a variety of geochemical and isotopic (δ13C) analyses to understand microbial methane cycling in an Athabasca oil sands region (AOSR) pit lake (Base Mine Lake) over a four-year period from 2015 to 2018. Distributions and δ13C values of phospholipid fatty acids (PLFAs) demonstrated that dissolved methane originating from underlying fluid fine tailings (FFT) was an important carbon source for the water column microbial community at all sampling times. Mass balance modelling results indicated that methane dissolution from bubbles was an important source of dissolved methane. Furthermore, the addition of alum to the lake in fall of 2016 had major impacts on microbial biogeochemistry.

Overall, this is a well-written manuscript that makes a significant contribution to our understanding of biogeochemical processes in oil sands pit lakes, which are considered to be one of the leading aquatic reclamation strategies in the AOSR. The discussion is supported by the data presented and the arguments are generally easy to follow. There are however several parts of the manuscript that require additional information and rearrangement of text. These instances and additional comments and suggestions are provided below:

 Response 1: We appreciate the reviewers positive comments.  We have addressed the reviewers suggestions as described below.

Abstract, line 23:

Define FFT

 Response 2: This change has been made

Line 24:

The second ‘throughout the water column’ is redundant and can be removed.

 Response 3: This change has been made

Line 52:

Concentrations of PAHs and alk-PAHs in OSPW are generally not very high. Naphthenic acids are typically the main organic contaminant of concern in tailings ponds.

 Response 4: We agree, we had made mention of the PAHs as we were highlighting the presence of these compounds.  But have removed the reference to the PAHs and only retained the reference to NA

Figure 1:

On the right-side figure, I suggest making the crossed-out circles more transparent so that the underlying text can be seen.

 Response 5: This change has been made

Figure 1 heading (lines 78-79):

What about the role of temperature in controlling the rates of methanotrophy / heterotrophy? Wouldn’t the rates be lower in winter due to lower water temperatures?

 Response 6: We have edited the figure heading to include reference to the potential slowdown of methanotrophic rates.

Line 178-180:

I find the description of parallel samples confusing. If 30 mL of water was used for CH4 concentrations, and 50 mL of water was used for isotopic analyses, what was the 60 mL of water that was collected used for?

 Response 7: The description of sampling has been edited to more clearly indicate that 30mL was collected for concentrations and 50 mL for isotopic analyses.  The samples were collected into 60 mL serum bottles, and the headspace that remained (30 and 10 mL respectively has been clarified).  No 60mL water samples were collected.

Line 191:

What was the volume of the headspace? 60 mL? Did the headspace volume vary considerably?

 Response 8: The FIS samples were 60 mL of FFT so the headspace was 60 mL in the 120 mL bottles used for these samples.  This has been more clearly stated in the text.

Line 206:

How much helium did you know to add to reach atmospheric pressure?

 Response 9: We have clarified that circa 25mL of He was added until pressure was equalized with atmosphere

Line 213:

How did you calculate aqueous CH4 concentrations from headspace concentrations? i.e., what proportion of CH4 did you assume was distributed into the headspace?

 Response 10: We have clarified on line 221 that all methane was assumed to have partitioned into the headspace, which is commonly assumed for methane concentration analysis and is particularly true given the initial sampling was into evacuated bottles..  As such methane concentrations were determined by dividing headspace methane mass into the original water sample volume.

Line 247:

Were the PVDF filters cleaned prior to use?

 Response 11: Yes the filters were rinsed with methanol.  This has been added to the text.

Line 255:

I assume the authors meant to say ‘isotopically characterized methanolic KOH’. Also, I suggest moving ‘δ13C’ to before KOH i.e., ‘isotopically characterized (δ13C) methanolic KOH’.

Also, I assume that the methanol in the 1:1 toluene:methanol mixture was isotopically characterised?

 Response 12: Yes, we apologize for the confusing structure.  This has been edited to read “… re-dissolved with KOH and a 1:1 toluene:methanol mixture (using methanol with known (δ13C) to clarify.

Line 277:

Should be ‘isotopically characterized methanolic KOH…”

 Response 13: This change has been made

Line 296:

What water temperatures did this model assume?

Response 14: Text has been expanded to clarify that the value used was representative of the highest concentrations observed in near surface FFT (at circa 1.5 m) and that the value was consistent with the model using in situ temperature of 14C.

 Line 359:

Should be Figure 3.

 Response 15: This change has been made

Somewhere below line 359:

Considering the data are shown in Figure 3, this section is missing a discussion on general trends in δ13C-CH4 values in the water column.

 Response 16: Thank you, this is a good point, this section must have been lost during the editing process. A description of the data has been re-inserted.

Line 364:

I suggest adding the following sub-section title: ‘Fluid Fine Tailings (FFT)’, or something similar.

 Response 17: This change has been made

As for the missing discussion on water column δ13C-CH4 values, this section needs a discussion on overall trends in δ13C-CH4 values from FFT.

 Response 18: Again, thank you, the section has been added.  Also included was a reference to the δ2H data presented in the SI.

Figure 4 heading (line 374):

Add the following to the end of the text: (see sampling site locations on Figure 2).

 Response 19: This change has been made

Line 398-399 (and throughout manuscript):

The terms 16:1 and 18:1 already indicate an unsaturated PLFA; thus the use of the suffix ‘unsat’ is redundant and can be removed.

 Response 20: This change has been made

Line 399:

For an AOSR-relevant study demonstrating significant 13C-depletion in 16:1 PLFAs as a result of methanotrophy (similar to that reported by the authors here), see Ahad et al., 2018, Science of the Total Environment.

 Response 21: We have added this observation to end of the paragraph discussing the isotopic depletion of the C16:1 PLFA.

Lines 411-412:

“While C18:1unsat PLFA are produced by a variety of organisms, they are often associated with phototrophic organisms.” References?

Response 22: We have cited the reference given earlier in the paragraph (Sakamoto et al 1994) and added Djikman 2010 to both locations to further clarify the support for this comment.

Line 431:

The 4-6‰ depletion in PLFAs compared to carbon source for heterotrophic bacteria in aerobic environments stated by the authors is a bit higher than that reported in the literature. In general, this difference is closer to 3‰. For instance, the paper cited by the authors (Hayes, 2001) states the following (page 23):

“A third approach was taken by Blair et al. (1985) who grew E. coli on glucose while obtaining a complete mass balance. They confirmed the finding of Monson and Hayes (1982a) that the fatty acids are depleted in 13C by 3‰ relative to the glucose carbon supply.”

Other investigations have also reported an approximate 3‰ rather than up to 6‰ depletion in PLFAs relative to carbon source (e.g., Boschker et al., 2005, Limnol & Oceanogr; Teece et al., 1999, Org. Geochem.), including one relevant study that examined microbial carbon sources in surface sediments from AOSR tailings ponds (Ahad et al., 2013, Environ. Sci. Technol.).

Response 23: We agree that the range we had cited was larger and more variable than can be applied.  We have updated the comment to reflect the 3 ‰ difference and added the comparison to the Ahad paper.

Line 441:

What exactly do the authors mean when they refer to the isotopic compositions of ‘pooled’ PLFAs? Do the authors mean to say that several peaks were integrated together using GC-IRMS software?

 Response 24: This was a comment that was apparently also lost during revision prior to submission.  A comment has been reinserted to the methods to clarify that yes, for peaks that were not felt to be sufficiently baseline separated, a pooled δ13C value was generated by integrating across several GC-IRMS peaks along with several references.

Line 464:

See comment above for line 431. AOSR bitumen has a δ13C value of around -30‰ (Ahad et al., 2013, Environ. Sci. Technol.); thus microbial utilisation of residual bitumen-derived hydrocarbons in FFT would be consistent with δ13C values close to -33‰ in PLFAs.

 Response 25: We have added a note to indicate this in the manuscript.

Line 500:

Should be Figure 3.

Response 26: This change has been made

Line 528:

What is the value of -61‰ based on? The most 13C-enriched δ13C-CH4 value found in FFT?

Response 27: Correct, the text has been edited to indicate this more clearly.

Lines 555-558:

Could lower rates of methane oxidation during Feb-May 2018 have also been caused by lower water temperatures?

Response 28: Yes this possibility has now been included in the text.

Line 648:

Should be -40‰.

 Response 29: This change has been made

Lines 682-685:

This sentence is awkward and needs to be reworded.

Response 30: The phase “that including markers” has been removed from the sentence to clarify the text.

Reviewer 3 Report

I liked the work, quite correct approaches were used to study the complex process of methane absorption by bacteria in a real polluted reservoir. At the same time, some results are quite expected - for example, a decrease in the activity of methane oxidants in winter. This is a reason for further research. 

Author Response

We appreciate the reviewers positive assessment of the manuscript.  

Reviewer 4 Report

The manuscript is devoted to the study of the methane cycle in the Base Mine Lake through the study of the chemical and isotopic (δ13C) compositions. The research is highly relevant and timely. The authors made important conclusions about the need for a systematic study of the dynamic nature of microbial communities and the influence of disturbances on biogeochemical cycles in lakes of oil sands open pit mines.

Overall, the manuscript is interesting and can be published in a journal.

The Introduction provides an overview of the current state of the problem, but here I would like to see a little more international articles on this topic. In the Materials and Methods  section the analytical methods are fully described. Results and Discussion fully reveal the main findings of the research. Conclusion reflect the main results of the research and fully reveal the intention and results of the future article.

My comments are listed below:

  1. Decipher the abbreviation FFT in Abstract
  2. Check all references to literature and arrange them in accordance with the requirements of the journal.
  3. The manuscript must be corrected according to the requirements of the journal. Now it contains a lot of inaccuracies: there is no numbering of subheadings, they contain different fonts, captions for figures and tables have different formatting, different spacing between paragraphs and sections, etc. This should be fixed.
  4. Perhaps it would be better if the Conclusion is made in the form of separate short, capacious conclusions. At the moment it looks like a continuation of the Discussion.

In my opinion the manuscript can be published after intensive error correction and careful checking references. It is also necessary to correct the manuscript according to journal requirements.

Author Response

The manuscript is devoted to the study of the methane cycle in the Base Mine Lake through the study of the chemical and isotopic (δ13C) compositions. The research is highly relevant and timely. The authors made important conclusions about the need for a systematic study of the dynamic nature of microbial communities and the influence of disturbances on biogeochemical cycles in lakes of oil sands open pit mines.

Overall, the manuscript is interesting and can be published in a journal.

The Introduction provides an overview of the current state of the problem, but here I would like to see a little more international articles on this topic. In the Materials and Methods  section the analytical methods are fully described. Results and Discussion fully reveal the main findings of the research. Conclusion reflect the main results of the research and fully reveal the intention and results of the future article.

 Response 1: We appreciate the reviewers opinion that the study is relevant and timely.  We have addressed the points the reviewer has raised to improve the manuscript as described below.  We were not quite clear whether there were specific international articles that the reviewer felt should be included in the introduction. We were trying to focus on early references but also recent studies in similar environments without having the number of references become excessive. We have added some references to the introduction, but if we have missed relevant articles we would certainly want to include them. 

My comments are listed below:

  1. Decipher the abbreviation FFT in Abstract

Response 2: This change has been made

  1. Check all references to literature and arrange them in accordance with the requirements of the journal.

Response 3: MDPI endnote style downloaded and used to format references

  1. The manuscript must be corrected according to the requirements of the journal. Now it contains a lot of inaccuracies: there is no numbering of subheadings, they contain different fonts, captions for figures and tables have different formatting, different spacing between paragraphs and sections, etc. This should be fixed.

Response 4: We have reviewed the instructions to authors from the website and perhaps they have recently changed, but we did not see a requirement for numbering of sections.  The issues with different fonts have been corrected.

  1. Perhaps it would be better if the Conclusion is made in the form of separate short, capacious conclusions. At the moment it looks like a continuation of the Discussion.

Response 5: We feel that the conclusions is a summative section highlighting the conclusions of the study rather than a continuation of the discussion.  As per the request of another reviewer, we have added some comments related to the implications of the study to lake management. 

In my opinion the manuscript can be published after intensive error correction and careful checking references. It is also necessary to correct the manuscript according to journal requirements.